# A Small Molecule Targeting Human MEK1/2 Enhances ERK and p38 Phosphorylation under Oxidative Stress or with Phenothiazines

**DOI:** 10.3390/life11040297

**Published:** 2021-03-31

**Authors:** Michał Otręba, Johanna Johansson Sjölander, Morten Grøtli, Per Sunnerhagen

**Affiliations:** 1Department of Drug Technology, Faculty of Pharmaceutical Sciences in Sosnowiec, Medical University of Silesia, Jednosci 8, 41-200 Sosnowiec, Poland; 2Department of Chemistry and Molecular Biology, University of Gothenburg, S-405 30 Gothenburg, Sweden; johanna.johansson.sjolander@cmb.gu.se (J.J.S.); grotli@chem.gu.se (M.G.)

**Keywords:** MAP kinase pathways, redox signalling, small molecule kinase modulator, phenothiazines

## Abstract

Small molecules are routinely used to inhibit protein kinases, but modulators capable of enhancing kinase activity are rare. We have previously shown that the small molecule INR119, designed as an inhibitor of MEK1/2, will enhance the activity of its fission yeast homologue, Wis1, under oxidative stress. To investigate the generality of these findings, we now study the effect of INR119 in human cells under similar conditions. Cells of the established breast cancer line MCF-7 were exposed to H_2_O_2_ or phenothiazines, alone or combined with INR119. In line with the previous results in fission yeast, the phosphorylation of the MAPKs ERK and p38 increased substantially more with the combination treatment than by H_2_O_2_ or phenothiazines, whereas INR119 alone did not affect phosphorylation. We also measured the mRNA levels of *TP53* and *BAX*, known to be affected by ERK and p38 activity. Similarly, the combination of INR119 and phenothiazines increased both mRNAs to higher levels than for phenothiazines alone. In conclusion, the mechanism of action of INR119 on its target protein kinase may be conserved between yeast and humans.

## 1. Introduction

Three-layered mitogen-activated protein kinase (MAPK) cascades are ubiquitous in eukaryotic intracellular signalling systems, being involved in responses to both internal and external agents, including stressors, cytokines, and growth hormones. They regulate processes such as cell growth and proliferation, differentiation, apoptosis, and inflammation [1]. MAPK cascades are attractive targets for the pharmacological treatment of a wide variety of disorders, including inflammatory diseases, diabetes, and cancer [1,2,3]. In the human p38 MAPK pathway, the MAPK kinases (MAPKKs) MKK3/4/6 phosphorylate and activate p38 under a variety of stress conditions including oxidative stress, cytokines, and osmotic shock. In the extracellular signal-regulated kinase (ERK) pathway, the MAPKKs MEK1/2 activate the MAPKs ERK1/2 in response to hormone and growth factor stimulation, but also to reactive oxygen species (ROS).

Chlorpromazine, fluphenazine, and thioridazine are from the phenothiazine class of drugs with antipsychotic properties that are used in the treatment of schizophrenia and bipolar disorders [4]. Moreover, research from the last 15 years suggests that phenothiazines could play an important role in the treatment of other diseases: cancer as well as viral, bacterial, and fungal infections [5,6,7,8]. The anticancer activity of phenothiazines is related in part to reactive oxygen species (ROS) production, DNA fragmentation, and stimulation of apoptosis as well as inhibition of migration and invasiveness [5,8,9,10]. Our previous studies using normal human epidermal melanocytes showed that chlorpromazine, fluphenazine, and thioridazine caused oxidative stress by increasing H_2_O_2_ levels and modulating the activity of the antioxidant enzymes superoxide dismutase (SOD), catalase (CAT), and glutathione peroxidases (GPx) [10,11,12,13].

INR119 is an analogue of PD98059, an allosteric inhibitor of MEK1 [14,15]. When tested in the fission yeast *Schizosaccharomyces pombe*, it was found that applying INR119 under oxidative stress enhanced signalling from the MAPKK Wis1, a structural homologue of MEK1 [16]. This enhancement was due at least in part to the prevention of oxidation of a cysteine residue, just upstream of the Asp-Phe-Gly motif in the kinase catalytic site. This cysteine is evolutionarily conserved among MAPKKs, including Wis1, MEK1/2, and MKK3/4/6.

INR119 was designed based on docking experiments using the structure of MEK1 bound to ATPMg in complex with the allosteric modulator PD0325901 (PDB 1S9J) [17]. These experiments suggested that INR119 formed three hydrogen bonds with the protein backbone; one between the chromone carbonyl oxygen and NH of Val 211 in the activation loop (C=O⋯H-N), a second hydrogen bond between the chromone carbonyl oxygen and NH of Ser212 in the activation loop (C=O⋯H-N), and a third between the aniline NH_2_ and the carbonyl oxygen of Phe 209 (N-H⋯O=C) [14]. The structure of INR 119 is shown in Figure 1.

We now wanted to see if we could extend our findings on the effects of INR119 on fission yeast Wis1 to mammalian cells. While selective small molecule inhibitors of protein kinases are numerous, very few protein kinase activators have been described. INR119 thus represents a rare opportunity to modulate a kinase signalling pathway through the upregulation of a specific protein kinase. The selective enhancement of the activity of a MAPK by pharmacological intervention during cancer therapy might enable us to, e.g., tip the response of the targeted cancer cell towards apoptosis. We chose to investigate the effect of the oxidative stress agent H_2_O_2_ or phenothiazines, alone or in combination with INR119, on the cultured human breast cancer cell line MCF-7. We studied the phosphorylation of the MAPKs ERK1/2, the target of the MAPKKs MEK1/2, and of the MAPK p38, the target of the MAPKKs MKK3/4/6. We further measured the levels of *TP53* and *BAX* mRNA, known to be regulated in part by ERK and p38. We found that the phosphorylation levels of both ERK and p38, which were elevated by H_2_O_2_ or phenothiazines, were further raised when these agents were combined with INR119. The combination of INR119 and phenothiazines also elevated *TP53* and *BAX* mRNA.

Our findings indicate a potential for small molecules such as INR119 or derivatives thereof to be used as adjuvants to potentiate the effect of, e.g., oxidizing anticancer drugs.

## 2. Materials and Methods

### 2.1. Cell Culture and Reagents

Human breast cancer MCF-7 cells were obtained from the ATCC and cultured in DMEM basal medium at 37 °C in 5% CO_2_. The medium was supplemented with foetal bovine serum (10%), amphotericin B (2.5 μg/mL), and Penicillin-Streptomycin (50 U/mL). Chlorpromazine hydrochloride, fluphenazine dihydrochloride, thioridazine hydrochloride, anisomycin, H_2_O_2_, DMSO, HEPES buffer pH 7.5, NaCl, Triton X-100, bromophenol, MgCl_2_, EDTA, Protease Inhibitor Cocktail, and PhosSTOP™ were purchased from Sigma-Aldrich (Saint Louis, MO, USA). Gibco™ BenchStable™ DMEM, Gibco™ Fetal Bovine Serum (qualified, heat-inactivated, United States), amphotericin B (250 µg/mL), Penicillin-Streptomycin (5000 U/mL), and StemPro^®^ Accutase^®^ Cell Dissociation Reagent were obtained from Thermo Fischer Scientific (Waltham, MA, USA). Tris(hydroxymethyl)aminomethane (TRIS, Trometamol) Electran^®^ for electrophoresis and dithiothreitol were obtained from VWR (Randor, PA, USA). Glycine was obtained from Alfa Aesar (Haver Hill, Fullerton, CA, USA), while sodium dodecyl sulfate was obtained from Merck (Darmstadt, Germany). INR119 was synthesized as previously described [14].

### 2.2. Treatment of Cells with INR119 and Oxidizing Agents

Cells were seeded in 6-well plates (Falcon) and incubated with basal medium up to about 80% confluence. Next, cells were incubated with H_2_O_2_ (1.0 mM) for 7.5, 15, or 30 min with and without INR119 (0.1 or 1.0 µM) for 30 min. Alternatively, cells were incubated for 30 min with chlorpromazine, fluphenazine, or thioridazine (1 or 5 μM each) with or without INR119 (1.0 µM). For mRNA analyses, cells were then washed with fresh medium and further incubated in normal medium at 37 °C for the indicated time periods. All experimental and control samples contained DMSO at a final concentration of 0.1% unless indicated otherwise.

### 2.3. Western Blot Analysis

The amounts of ERK1/2, phospho-ERK1/2, MEK1/2, and phospho-MEK1/2 were determined by western blot analysis according to standard methods [18]. Cells were lysed directly in ice-cold lysis buffer at 4 °C for 30 min with gentle agitation. Cell lysis buffer contained 50 mM HEPES buffer pH 7.5, 10 mM NaCl, 1% Triton X-100, 10% glycerol, 5 mM MgCl_2_, 1 mM EDTA, Protease Inhibitor Cocktail, PhosSTOP™, and distilled water. The protein concentrations were analysed by using a Pierce BCA Protein Assay Kit (Thermo Fischer Scientific, Waltham, MA, USA). Nonused samples were stored at −80 °C for a qPCR analysis. Samples were diluted in SDS-PAGE sample loading buffer containing 0.5 M Tris/HCl pH 6.8, 40% glycerol, 10% SDS, bromophenol, distilled water, and 100 mM dithiothreitol, and were heated at 95 °C for 5 min. Proteins were separated on 10% SDS-PAGE along with protein markers and were transferred onto nitrocellulose membranes using a semidry blotting machine (Bio-Rad., Hercules, CA, USA). The membranes were blocked in blocking buffer [5% (*w/v*) nonfat dried milk in TBST buffer (0.05% (*v/v*) Tween 20/TBS)] for 1 h at room temperature. Proteins were detected by incubation with primary antibodies, all from Cell Signaling Technology: Phospho-p44/42 MAPK (#4370), Phospho-p38 MAPK (#9216), p44/42 MAPK (#9212), and p38 MAPK (#4695) diluted at 1:2000, 1:500, or 1:1000, respectively, in blocking buffer overnight at 4 °C. The membranes were washed with TBST solution and then incubated at room temperature using a cradle shaker with secondary peroxidase antibodies, both from Sigma-Aldrich: goat anti-mouse IgG whole molecule or goat anti-rabbit IgG whole molecule, both diluted at 1:2500. Immunoreactive bands were visualized using a Clarity Western ECL substrate (Bio-Rad) following the manufacturer’s protocol. The signals were detected with Bio-Rad ChemDoc MP Imaging System (Bio-Rad) and expressed as percentages of the controls.

### 2.4. Quantitative Real-Time PCR (qPCR)

Total mRNA was extracted from the lysates used for the western blot using a GenElute Mammalian Total RNA Miniprep Kit (Sigma Aldrich) following the manufacturer’s instructions. Genomic DNA was eliminated by treatment with DNase I (Thermo Scientific) at 37 °C for 30 min, which was subsequently inactivated by incubation with 0.1 M EDTA for 10 min at 65 °C. The cDNA was synthesized by reverse transcription using Oligo dT Thermo Scientific and dNTPs (10 mM each) (Thermo Scientific, Waltham, MA, USA) as well as 0.1 M DTT, 5X First-Strand Buffer, and M-MLV reverse transcriptase 40,000 u (200 u/μl) (Invitrogen, Carlsbad, CA, USA) according to the manufacturer’s protocol. For the real-time PCR, gene-specific primers were obtained from Invitrogen: TP53 forward (5′-GCATTTGCACCTACCTCACA-3′), reverse (5′-GTGGTTTCAAGGCCAGATGT-3′); BAX forward (5′-ACTTTGCCAGCAAACTGGTG-3′), reverse (5′-CAGCCCATGATGGTTCTGAT-3′); and ACTB forward (5′-AAACTGGAACGGTGAAGGTG-3′), reverse (5′-CTCGGCCACATTGTGAACTT-3′). The real-time PCR was performed using a BioRad CFX Connect Real-Time System and 10 µL of a solution containing 2 µL of cDNA, 0.5 µL of forward and reverse primers (10 pmol/µL each), and 2 µL of 5 × HOT Fire Pole Eva Green qPCR Mix (no ROX) (Solis BioDyne) for each reaction. Next, the 96-well plate was centrifuged for 1 min, 1000 rpm at room temperature. The qPCR reaction conditions were as follows: one cycle of initial denaturation at 95 °C for 15 min, 40 cycles of denaturation at 95 °C for 10 s, annealing at 60 °C for 20 s, and elongation at 72 °C for 20 s. The levels of *TP53* and *BAX* mRNA were normalized to *ACTB* (actin B) expression. The real-time PCR data were analysed by calculating the 2^−ΔΔCt^ value for each experimental sample.

### 2.5. Statistical Analysis

In the western blot and qPCR analyses, the mean values of at least three separate experiments (n = 3) ± standard deviation (SD) were calculated. A statistical analysis was performed with a one-way ANOVA (comparison samples vs. control) with Dunnett’s multiple comparison test using GraphPad Prism 8 software. The significance level was established at *p* < 0.05 (*) or *p* < 0.01 (**).

## 3. Results

### 3.1. Phosphorylation of ERK1/2 and p38 under H_2_O_2_-Induced Oxidative Stress Is Enhanced by INR119

We exposed cultured MCF-7 cells to 1 mM H_2_O_2_ for up to 30 min and observed the change in ERK1/2 or p38 phosphorylation (Figure 2). Anisomycin was included as a positive control for MEK1/2 or MKK3/4/6 activation. In line with previous observations in cultured mammalian cells with comparable concentrations of H_2_O_2_ [19,20,21], there was a minor increase in ERK1/2 phosphorylation during this time period (Figure 2A) but no detectable increase of p38 phosphorylation (Figure 2B). In fission yeast, we previously observed an enhanced phosphorylation of the MAPK Sty1 by INR119 with a maximum at around 30 min of exposure to 0.5 mM H_2_O_2_ [16]. The EC_50_ of INR119 in MCF-7 cells with respect to ERK1/2 phosphorylation is <1 μM [16]. The addition of INR119 to MCF-7 cells at 0.1 μM in the presence of 1 mM H_2_O_2_ caused ERK1/2 phosphorylation, as measured at 90 min after the experiment onset, to increase robustly up to 30 min of exposure, and with 1 μM of INR119 the increase was further increased at 15 and 30 min to about 2-fold higher than with H_2_O_2_ only (Figure 2A). For p38, the enhancement by INR119 was stronger. While H_2_O_2_ only did not increase p38 phosphorylation, 1 μM of INR119 caused it to double at 15 min and to increase ~5-fold at 30 min of exposure (Figure 2B). For cells without H_2_O_2_, after 30 min of exposure to INR119 at 0.1 μM, ERK1/2 and p38 phosphorylation decreased to about half their original value, but at 1 μM of INR119 phosphorylation increased 2-fold for ERK1/2 and 2.3-fold for p38 (Figure 2D). Hence, the combination of 1 μM INR119 and 1 mM H_2_O_2_ resulted in p38 phosphorylation increasing 2.5-fold over INR119 only and 5-fold over H_2_O_2_ only. For ERK1/2 phosphorylation, the corresponding values were 3-fold and 2-fold, respectively. Thus, for the phosphorylation of both MAPKs, there is a synergism between INR119 and H_2_O_2_.

### 3.2. Phosphorylation of ERK1/2 and p38 under Phenothiazine-Induced Oxidative Stress Is Enhanced by INR119

The phenothiazines chlorpromazine, fluphenazine, and thioridazine have been shown to cause intracellular oxidative stress [9,10,11]. We wanted to see if these pharmaceuticals could likewise increase the phosphorylation of ERK1/2 and p38 in conjunction with INR119, and we exposed cells to 1 or 5 μM of each phenothiazine for 12 h, or 5 μM for 24 h, in the presence or absence of 1 μM of INR119.

After 12 h, phenothiazines at 1 μM caused a moderate decrease of ERK1/2 and p38 phosphorylation, with or without INR119 (Appendix A); at 5 μM of phenothiazines, the phosphorylation levels were indistinguishable from untreated cells (Appendix A). After 24 h with 5 μM of phenothiazines, the phosphorylation of ERK1/2 increased slightly in the presence of INR119 for fluphenazine and of p38 for thioridazine. At these timepoints, INR119 alone did not impact the phosphorylation of either MAPK (Figure 3).

### 3.3. TP53 and BAX Expression under Phenothiazine-Induced Oxidative Stress Is Enhanced by INR119

To investigate the effects on signalling downstream of ERK and p38, we analysed the expression of two downstream genes, *TP53* and *BAX*, partly regulated by ERK and p38 [22,23,24,25], treated with phenothiazines at 5 µM for 24 h with and without treatment for 30 min with INR119 (1 µM). Under these conditions, all three phenothiazines alone suppressed the expression of both *TP53* and *BAX*. The combination of phenothiazines and INR119, however, enhanced the transcript levels of *TP53* and *BAX* 2–4-fold (Figure 4). For both transcripts, fluphenazine had the strongest impact. In contrast to the lack of effect of INR119 alone on the phosphorylation of ERK1/2 and p38 at these timepoints (Figure 3), it did increase the levels of both transcripts. This indicates that, in the absence of any oxidative stress agent, INR119 activates another unidentified pathway beside ERK and p38 and that that pathway is responsible for the transcriptional induction of *TP53* and *BAX*. Sorbitol at 0.5 M, the positive control for MEK1/2 or MKK3/4/6 activation, increased the *BAX* but not *TP53* levels (Figure 4).

## 4. Discussion

Here, we show that in the presence of oxidative stress, from H_2_O_2_ or phenothiazines, INR119 causes the enhancement of ERK1/2 and p38 phosphorylation. This parallels our findings in the fission yeast *S. pombe*, where INR119 targets the stress-activated MAPK Wis1. In unstressed *S. pombe*, INR119 depresses the phosphorylation of the MAPK Sty1 by Wis1, whereas under H_2_O_2_-induced oxidative stress, INR119 instead enhances the phosphorylation activity of Wis1 towards Sty1 [16]. INR119 can thus be considered a conditional protein kinase modulator.

In fission yeast, genetic evidence showed that the INR119-dependent enhancement of phosphorylation of the MAPK Sty1 was completely dependent on its sole MAPKK, Wis1, and on the presence of at least one of its dedicated upstream MAPKKKs, Win1 and Wis4 [16]. This indicates that the effect is indeed channelled through the canonical three-tiered MAPK cascade. INR119 was designed to target an allosteric site of MEK1/2 [14,15], close to a Cys residue just upstream of the Asp-Phe-Gly triad that is conserved in all members of the eukaryotic Ser/Thr/Tyr protein kinase family. This Cys residue is common to many MAPKKs of yeast and human origin but is not found in other branches of the protein kinase sequence tree [16]. We showed that the enhancement of Sty1 phosphorylation under oxidative stress by INR119 was dependent on this residue (Wis1 Cys458), as expected if the small molecule indeed targets the MAPKK by binding in its vicinity, which is also in agreement with molecular modelling data [16]. Thus, we predict that the action of INR119 on MEK1/2 and MKK3/4/6 will analogously be mediated by the cognate conserved cysteine residue in those MAPKKs.

In the presence of 1 mM H_2_O_2_, 1 µM INR119 enhanced the phosphorylation of both MAPKs. The most prominent effect was seen for p38, and in a somewhat weaker way for ERK1/2. INR119 at 1 µM produced a stronger phosphorylation enhancement than at 0.1 µM. There was a steady increase in the time interval of 7.5 to 30 min (Figure 2). The combination of INR119 and H_2_O_2_ caused a phosphorylation enhancement up to 5-fold higher than for H_2_O_2_ alone. This combination also caused an enhancement up to 2.5-fold higher than for INR119 alone. These results are consistent with the time course and amplitude of phosphorylation enhancement seen in fission yeast [16]. This demonstrates that oxidative stress, in this case by peroxides, is sufficient for INR119 to robustly activate MAPKKs in mammalian cells as well as in fission yeast.

For the combinations of INR119 and phenothiazines, we observed only a moderate enhancement of MAPK phosphorylation. The phenothiazines tested here have been shown to affect oxidative stress in mammalian cells. Chlorpromazine increased SOD activity and increased H_2_O_2_ content at concentrations below 1.0 µM, and increased GPx activity at 0.1 µM, while chlorpromazine or fluphenazine treatment at 0.1 and 1.0 µM decreased CAT activity [12,13]. Thioridazine from 0.01 to 2.5 μM likewise increased H_2_O_2_ content. This drug also affected enzymes in the redox metabolism; between 0.1 and 2.5 μM, SOD activity increased while CAT activity decreased. Thioridazine at 0.1 µM increased GPx activity but decreased it at higher concentrations [10]. These observations indicate a complex relationship between the drug dosage and intracellular redox status. In this work, the phenothiazines by themselves at 5 µM caused a moderate decrease of ERK and p38 phosphorylation (Figure 3 and Appendix A). In combination with INR119, however, there was a moderate enhancement of MAPK phosphorylation. Reactive oxygen species generated by phenothiazines, together with INR119, may activate MEK1/2 and MKK3/4/6 enough to counteract the inhibitory effect of phenothiazines on MAPK phosphorylation and turn the net effect into enhanced MAPK phosphorylation. This would enhance the cancer cell killing and proapoptotic effect of phenothiazines. Fluphenazine decreased the viability of 4T1 and MDA-MB-231 breast cancer cell lines while decreasing the expression of p44/42 ERK and increasing the ROS level in both cell lines [11]. This may be at least partly due to apoptosis, as ERK 1/2 signalling positively regulates the expression of proapoptotic proteins including BAD, BAX, and NOXA [26].

Regarding the effects of INR19 on gene expression, INR119 on its own did not affect ERK or p38 phosphorylation under the conditions and exposure times (12 or 24 h) studied here (Figure 3 and Appendix A). Nevertheless, the mRNA levels of both *TP53* and *BAX* were enhanced by the single treatment with INR119 (Figure 4). This can only be explained by INR119 activating other, as yet unknown, signalling pathways beside MEK1/2–ERK1/2 and MKK3/4/6–p38 in mammalian cells. In the parallel case in fission yeast, we previously observed that the hyperactivation by INR119 of the downstream target gene *srx1^+^* was partially, but not completely, dependent on the homologous Wis1–Sty1 pathway [16].

In summary, our results show that a conditional modulator of protein kinase activity such as INR119 may use the ROS generated by ERK signalling to enhance its kinase activity and increase the expression of proapoptotic genes such as *TP53* and *BAX*, further elevating the cancer-cell killing effect. The enhancing effect on p38 and ERK1/2 phosphorylation was approximately the same, indicating that INR119 may affect MEK1/2 and MKK3/4/6 equally. Achieving selectivity for either of these MAPKK classes will require further refinements of INR119 with conventional structure-based approaches. Our work demonstrates how one aspect of the anticancer activity of phenothiazines can be improved.

## Figures and Tables

**Figure 1 life-11-00297-f001:**
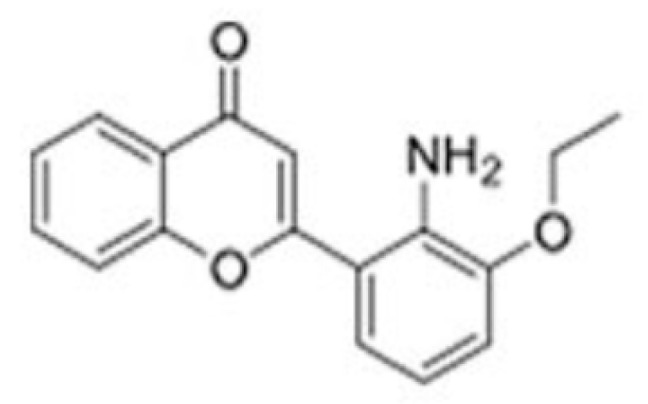
Structure of INR 119.

**Figure 2 life-11-00297-f002:**
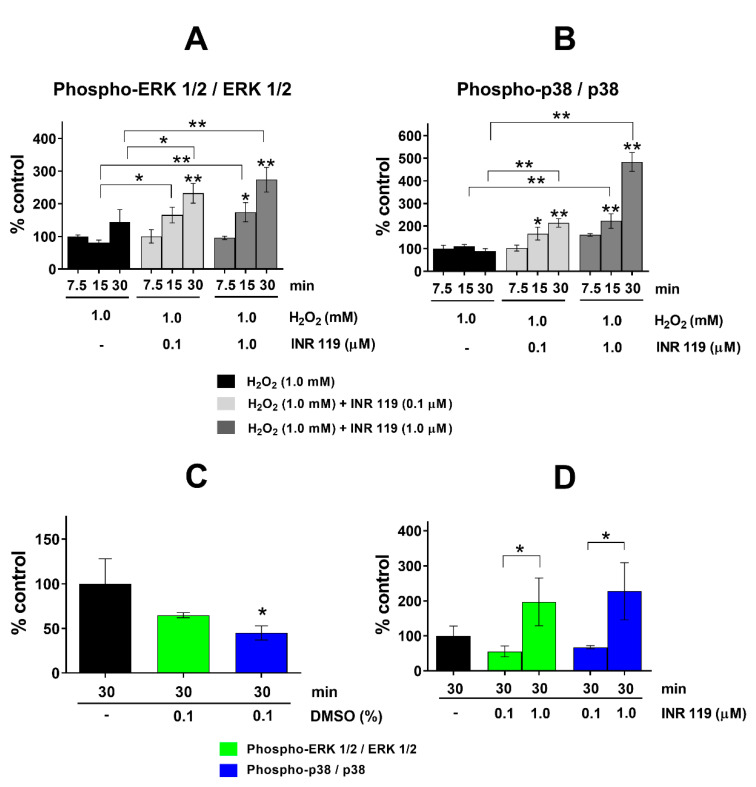
Phosphorylation of ERK1/2 and p38 under H_2_O_2_-induced oxidative stress. (**A**) Effect of incubation with H_2_O_2_ and INR119 (0.1 or 1.0 µM) on ERK1/2 phosphorylation. (**B**) Effect of the incubation with H_2_O_2_ and INR119 (0.1 or 1.0 µM) on p38 phosphorylation. (**C**) Impact of DMSO on the phosphorylation of ERK1/2 and p38. (**D**) Impact of INR119 on the phosphorylation of ERK1/2 and p38. Error bars show the standard deviation (* *p* < 0.05, ** *p* < 0.01). The original sample blot corresponding to this figure is shown in Appendix A.

**Figure 3 life-11-00297-f003:**
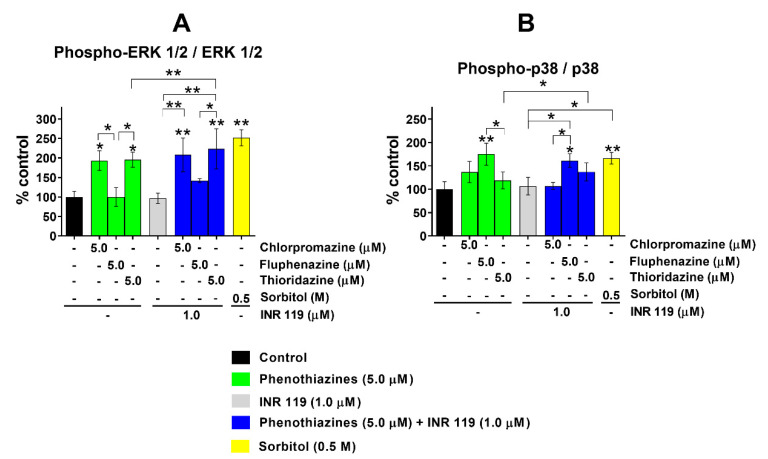
Phosphorylation of ERK1/2 and p38 under phenothiazine-induced oxidative stress after 24 h. (**A**) Impact of phenothiazines with and without INR119 on ERK1/2 phosphorylation. (**B**) Impact of phenothiazines with and without INR119 on p38 phosphorylation. Error bars show the standard deviation (* *p* < 0.05, ** *p* < 0.01). The original sample blot is shown in Appendix A.

**Figure 4 life-11-00297-f004:**
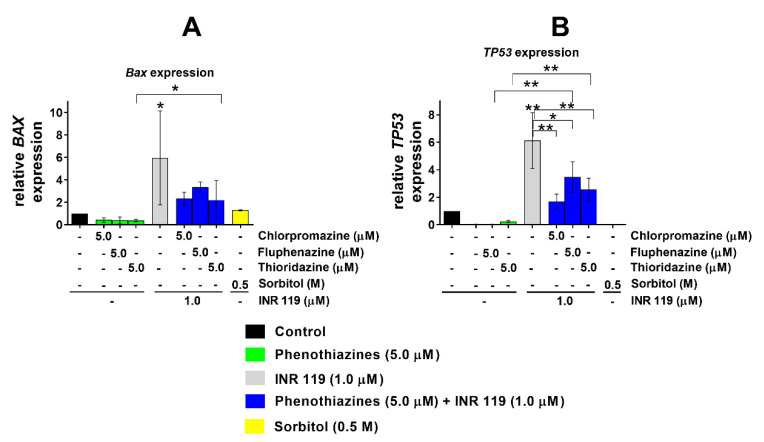
Expression of *TP53* and *BAX* mRNA (ΔΔC_t_ analysis) normalised to *ACTB* after 24 h treatment. (**A**) BAX expression relative to control (medium + DMSO 0.1%). (**B**) *TP53* expression relative to control (medium + DMSO 0.1%). Error bars show the standard deviation (* *p* < 0.05, ** *p* < 0.01). The raw qPCR data represented in this figure is found in Appendix A.

## Data Availability

The PCR data and Supplementary Figures presented in this study are available in Appendix A: https://zenodo.org/record/4559690 (accessed on 31 March 2021), DOI: 10.5281/zenodo.4559690.

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
