# Peer review of "A Small Molecule Targeting Human MEK1/2 Enhances ERK and p38 Phosphorylation under Oxidative Stress or with Phenothiazines"

_life, 2021, doi:10.3390/life11040297_

Round 1

Reviewer 1 Report

The corresponding authors submitted their study of how the lead compound INR119 enhancing the phosphorylation of kinases in the presence of oxidative stress. They got a tip from their earlier studies of developing INR119 as an inhibitor to MEK1/2, where they found it also enhances the activity of Wis1. In the present article, they tested those results on human cells.

However this article is poorly written with some ambiguous results. Here are some of the comments.

1) I think its important to show the structure of INR119 in the main article.

2) Format the figures

3) Figure headings are confusing: eg: In fig 1 A: The tile says Ohospho-p44/42MAPK/p44/42 MAPK, but in the discussion they are using the ERK1/2 to analyze the results. So, it would be good if they add ERK1/2 to the title.

4) Line 184, change Fig. 1B to 1D.

5) Lines 233-236: INR119 alone itself increased the phosphorylation by 6 folds, it is important to discuss about the probable reasons.

6) Rephrase the lines 265-270.

With the above corrections and proper editing of the paper, it is acceptable to publish

Reviewer 2 Report

This is an interesting article that follows previous reports on the interesting kinase ligand INR119. The effect on ERK and p38MAPK under the oxidative stress or exposure to the phenothiazine class of neurotropic drugs is described. The effects are of sufficient broad interest for publication. The authors provide a series of experiments that test explicit hypotheses about mechanism of action. The data are well summarized. Minor criticisms include lack of clarity in some figure legends, such as ones where the colors used in the figure are not readily presented in the legend to help the reader. A moderate scientific criticism is the unfinished nature of the work, including dose dependent effects of each compound alone or the usual drug-drug interaction screens that are standard in studies of tumor biology cell lines. However,it seems there is a lack of documentation on the MCF-7 line used, making further evaluation of compound synergy not the best use of time and effort for this cell line. The authors are encouraged to consider a set of ATCC registered tumor cell lines to better define the synergy and gain insight into the mechanisms involved. At this time, a short note is worth putting into the public domain.
